# Expression of PD-1, PD-L1 and PD-L2 in Lymphomas in Patients with Pre-Existing Rheumatic Diseases—A Possible Association with High Rheumatoid Arthritis Disease Activity

**DOI:** 10.3390/cancers14061509

**Published:** 2022-03-15

**Authors:** Erik Hellbacher, Christer Sundström, Daniel Molin, Eva Baecklund, Peter Hollander

**Affiliations:** 1Department of Medical Sciences, Rheumatology, Uppsala University, S-751 85 Uppsala, Sweden; eva.baecklund@medsci.uu.se; 2Clinical and Experimental Pathology, Department of Immunology, Genetics and Pathology, Uppsala University, SE-752 36 Uppsala, Sweden; christer.sundstrom@igp.uu.se (C.S.); peter.hollander@igp.uu.se (P.H.); 3Experimental and Clinical Oncology, Department of Immunology, Genetics and Pathology, Uppsala University, SE-752 36 Uppsala, Sweden; daniel.molin@igp.uu.se

**Keywords:** lymphoma, PD-1, PD-L1, rheumatic disease, immunohistochemistry

## Abstract

**Simple Summary:**

Immunotherapy blocking programmed cell death protein 1 (PD-1) and its ligands (PD-L1, PD-L2) is less effective in non-Hodgkin lymphoma (NHL) than classical Hodgkin lymphoma. However, NHL is a heterogeneous group and current research seeks to identify subgroups of NHL patients responsive to PD-1 blocking agents. Whether patients with pre-existing rheumatic diseases might constitute such a subgroup is unknown. We investigated the expression of PD-1 and its ligands in lymphoma patients with pre-existing rheumatic diseases. Our key findings include that in patients with pre-existing rheumatoid arthritis (RA) and subsequent diffuse large B-cell lymphoma, an association between RA disease severity and increased expression of PD-L1 in tumor cells was seen. This warrants further studies of the PD-1 pathway in lymphoma in other chronic inflammatory conditions.

**Abstract:**

Current research seeks to identify subgroups of non-Hodgkin lymphoma (NHL) patients responsive to PD-1 blocking agents. Whether patients with pre-existing rheumatic diseases might constitute such a subgroup is unknown. We determined intratumoral expression of PD-1 and its ligands in lymphoma patients with pre-existing rheumatic diseases. We included 215 patients with rheumatoid arthritis (RA), systemic lupus erythematosus (SLE) or Sjögren’s syndrome with subsequent lymphoma and 74 diffuse large B-cell lymphoma (DLBCL) controls without rheumatic disease. PD-1 and PD-ligand immunohistochemical markers were applied on tumor tissue microarrays. The number of PD-1+ tumor infiltrating leukocytes (TILs) and proportions of PD-L1+ and PD-L2+ tumor cells and TILs were calculated and correlated with clinical data. Expression of PD-L1 in tumor cells and TILs was highest in classical Hodgkin lymphoma and DLBCL. In DLBCLs, expression of PD-1 in TILs and PD-L1 in tumor cells was similar in RA, SLE and controls. In RA-DLBCL, high expression of PD-L1 in tumor cells was significantly more common in patients with the most severe RA disease and was associated with inferior overall survival in multivariable analysis.

## 1. Introduction

The inhibitory immunoreceptor programmed cell death protein 1 (PD-1) and its ligands PD-L1 and PD-L2 play crucial roles in maintaining immunological homeostasis, self-tolerance and preventing autoimmunity, mainly by suppressing T-cell activity. Many malignancies, including lymphomas, can exploit the PD-1 pathway by expressing PD-L1, leading to evasion of host anti-tumor immune response. In classical Hodgkin lymphoma (cHL), PD-L1 is commonly overexpressed by Hodgkin and Reed–Sternberg (HRS) cells [1], and increased PD-1+ tumor infiltrating leukocytes (TILs) [2,3,4] and PD-L1 expression on HRS cells [3,5] have been associated with worse outcome. Anti-PD-1 therapy is effective in relapsed or refractory (r/r) cHL [6,7]. With the exception of r/r primary mediastinal large B-cell lymphoma [8], various subtypes of non-Hodgkin lymphoma (NHL), including diffuse large B-cell lymphoma (DLBCL), have shown to be less sensitive to anti-PD-1 therapy [9,10]. However, NHL is a heterogeneous group and expressions of PD-1 and PD-L1 in tumor cells and cells in the tumor microenvironment (TME) vary across different NHLs as well as within specific lymphoma subtypes, potentially influencing the response to anti-PD-1 therapy.

Several rheumatic conditions, such as rheumatoid arthritis (RA), systemic lupus erythematosus (SLE) and primary Sjögrens’s syndrome (pSS), are associated with an increased risk of lymphoma [11,12,13,14,15,16]. There is mounting evidence of an important role for the PD-1 pathway in the development of autoimmune diseases [17,18,19,20]. Proinflammatory cytokines commonly overexpressed in rheumatic diseases [21,22,23] are also involved in regulation of expression of PD-1 and its ligands [24]. Additionally, the occurrence of immune related adverse events due to anti-PD-1 therapy, including rheumatic diseases such as RA, SLE and pSS [25], further emphasizes a link between the PD-1 pathway and rheumatic diseases. Therefore, it could be suspected that expression and/or function of PD-1 and its ligands in tumor cells and TILs in lymphoma differ between patients with and without pre-existing rheumatic diseases. Expression of PD-1, PD-L1 and PD-L2 has not previously been studied in a cohort of lymphoma patients with pre-existing rheumatic diseases. 

In this study, we assessed the expression of PD-1, PD-L1 and PD-L2 in lymphoma tissue from 215 patients with pre-existing RA, SLE or pSS, and in 74 DLBCL controls without rheumatic disease for comparison. Furthermore, we related expression of PD-1, PD-L1 and PD-L2 to clinical characteristics and survival outcome in RA patients with DLBCL. 

## 2. Materials and Methods

### 2.1. Identification of Study Subjects

Cases with RA or SLE and lymphoma were identified within previous population-based studies of risk factors for lymphoma [26,27,28]. All individuals with RA diagnosed in 1964–1994 (*n* = 74,651) or SLE diagnosed in 1964–1995 (*n* = 6438) were identified from the Swedish Inpatient Register. The following International Classification of Diseases [ICD] codes were used: RA [ICD-7]: 722, ICD-8: 712.38–712.39, ICD-9: 714A–C,714 W), SLE [ICD-7]: 456,20; ICD-8: 734,10; ICD-9: 710A). 

Through linkage with the Swedish Cancer Register (1964–2005 for RA, 1964–1995 for SLE), to which reporting is mandatory [29], individuals diagnosed with lymphoma ([ICD-7]: 200-202) as their first primary cancer after the first discharge listing RA or SLE were identified. The rheumatic diagnosis was verified by review of medical records and all lymphoma biopsies were reviewed and classified according to the WHO classification [30] by an experienced haematopathologist (CS). Cases diagnosed at autopsy were excluded. Of 343 verified RA-lymphoma cases, 188 cases with sufficient clinical data and lymphoma tissue for tissue microarray (TMA) construction were included in the study and of 26 verified SLE-lymphoma cases, 18 cases with sufficient tissue were included (Figure 1).

Patients with pSS were identified from consecutive patients fulfilling criteria for pSS diagnosed and followed-up at the Rheumatology department at Uppsala University Hospital, Sweden, during 1991–2002. Nine patients who developed lymphoma during this period had tumor tissue for TMA construction and were included in the study.

We used DLBCL patients without rheumatic disease from a previous population-based DLBCL study in Uppsala Health Care Region, Sweden, diagnosed in 1984–2002, as controls for comparison with the RA patients with DLBCL (RA-DLBCL)and the SLE patients with DLBCL (SLE-DLBCL) [31]. Specimens suitable for TMA construction remained in 74 cases. The study, complying with the Helsinki declaration, was approved by the Regional Ethical Review Board, Uppsala, Sweden (reference number 2006/350).

### 2.2. Clinical Information

Detailed clinical information of both the rheumatic disease and lymphoma was collected from medical records in previous studies [26,27,28]. For RA patients, disease activity had been individually scored repeatedly over time based on swollen and tender joint counts, erythrocyte sedimentation rate (ESR), and the treating physician’s global assessment of disease activity from onset of RA until lymphoma diagnosis, as described in detail previously [26]. Values of cumulative disease activity for each patient were calculated as the area under the curve (AUC). In order to investigate a possible impact of RA disease severity on the expression of PD-1 and its ligands, RA patients with DLBCL were divided into two groups depending on their AUC values, using the upper quartile as the cut-off for the highest disease activity group. Active treatment for the rheumatic disease was defined as the treatment at the time of lymphoma diagnosis, provided it had been used for four or more consecutive weeks. 

### 2.3. Lymphoma Subtyping and Analysis of Epstein–Barr Virus Status

DLBCLs were classified as either germinal center B-cell-like (GCB) or non-GCB according to the Hans algorithm [32]. The presence of Epstein–Barr virus (EBV) in lymphoma tissue was analysed using EBV-encoded RNA (EBER) in situ hybridization [33]. 

### 2.4. Tissue Samples 

Formalin-fixed, paraffin-embedded (FFPE) diagnostic biopsies were gathered and TMAs containing two cores of 1 mm in diameter per case were constructed using standard techniques [26,31]. 

### 2.5. Immunohistochemical Stainings

To identify PD-1, mouse monoclonal antibody (mAb) NAT105/ab52587 (Abcam, Cambridge, UK) was used, citrate buffer was used to retrieve the antigen and positive signals were envisioned with MACH3 mouse HRP reagents (brown) (Biocare Medical, Walnut Creek, CA, USA). Rabbit mAb E1L3N/13684 (Cell Signaling Technology, Danvers, MA, USA) was used to identify PD-L1 and rabbit mAb D7U8C/82723 (Cell Signaling Technology) was used to identify PD-L2. As described in previous studies, E1L3N/13684 [34] and D7U8C/82723 [35] detect membranous expression of tumor cells and immune cells, respectively. In order to accurately distinguish between PD-ligands expressed on tumor cells or TILs, double immunostaining for PD-ligands and the B-cell biomarker PAX-5 was performed. Mouse mAb M7307/DAK-Pax5 (Dako, Santa Clara, CA, USA) was used to identify PAX-5. Antigens for PD-L1, PD-L2 and PAX-5 were retrieved in Tris-EDTA buffer (pH 9.0) in a pressure cooker. Positive signals for PD-L1 and PD-L2 were envisioned with the Betazoid DAB detection kit (brown) (Biocare Medical), and PAX-5 was detected using Warp Red chromogen (red) (Biocare Medical). Lastly, the sections were counterstained with Intellipath FLX hematoxylin. All antibodies were diluted 1:50.

### 2.6. Evaluation of Immunohistochemical Stainings

The TMA slides were scanned in high-power magnification (×400). Each TMA core was 1 mm in diameter and consisted of an area of 0.785 mm^2^ (radius*radius*π=area) (0.5*0.5*π=0.785) and the total area for 2 cores was 1.57 mm^2^. The scanned slides were viewed and analyzed with the Visiomorph software (Visiopharm^®^, Hørsholm, Denmark). Different image analysis application algorithms were constructed for PD-1, PD-L1 and PD-L2. First, representative positive and negative areas were marked manually to train each application. The applications were trained to designate cells with a brown membranous staining of all intensities (PD-1, PD-L1 or PD-L2) as positive, while cells without brown membranous staining were designated as negative. In addition, for PD-L1 and PD-L2, the software was manually trained to designate cells with a red nuclear staining with PAX-5 as tumor cells, and PAX-5 negative cells as leukocytes. Next, using a TMA workflow application with each image analysis application applied, the software calculated the number of positive and negative cells for each application and calculated numbers and proportions of positive cells. The numbers and proportions for each marker were calculated as an average count of the two cores from that case. Areas with fibrosis, artifacts or reactive germinal centers were manually excluded from the analyses. Proportions of PD-L1 or PD-L2 positive leukocytes was calculated as the number of positive leukocytes divided by the total number of leukocytes (positive and negative). Proportions of PD-L1 or PD-L2 positive tumor cells was calculated as the number of positive tumor cells divided by the total number of tumor cells (positive and negative). In order to present results comparable with manual evaluations with a microscope, we present the number of PD-1 positive cells per 400× high power field (HPF) (0.0625 mm^2^). The ratio between the area of the 2 cores and the area of 1 HPF was calculated (1.57/0.0625=25.13). Thus, to calculate the number of PD-1 positive cells per HPF, the number of positive cells in the 2 cores were divided by 25.13 for each case. In order to compare our findings with previous studies on cHL [3], PD-1 in TILs was also calculated as a proportion of positive cells in patients with cHL. On morphological examination, PD-1 was mainly expressed by lymphocytes (Figure 2G). 

### 2.7. Statistical Analysis

Optimal cut-off values for high and low expression of PD-1 in TILs and PD-L1 and PD-L2 in tumor cells and TILs were determined by receiver operating characteristic (ROC) curves with Yoden´s index calculated for each marker with dead at 12 months as the outcome in RA-DLBCL patients. With this approach, the following cut-off values for high proportion were used: ≥21 PD-1 positive cells/HPF for TILs, ≥17% PD-L1+ and ≥8% PD-L2+ for tumor cells, and ≥10% PD-L1+ and ≥2% PD-L2+ for TILs in patients with DLBCL (Figure 2). Overall survival (OS) was defined as time from lymphoma diagnosis to the time of last follow up or death of any cause. All patients were followed up for survival until death or 28 August 2020. Survival curves and univariate survival analyses were performed using the Kaplan–Meier method, log-rank test and Cox proportional hazards regression to compare differences between groups. Variables of statistical significance (*p* < 0.05) in the univariate survival analyses were included in the multivariable Cox regression model. The proportional hazards assumption was tested and was not validated. Tabulated values were compared using the Chi square or Fisher’s exact test. Wilcoxon signed rank-test was used to compare non-tabulated values between groups. Continuous variables were compared using the Spearman Rank Order Correlation test. A *p*-value of <0.05 was considered significant. Statistical analysis was performed using Rstudio 1.3.1093 and R with the Rcmdr package (R version 3.6.2).

## 3. Results

### 3.1. Patient Characteristics

Of the 215 included cases with a diagnosis of rheumatic disease before lymphoma development, 188 (87%) had RA, 18 (8%) SLE and 9 (4%) pSS (Table 1). Of these, 62% were women, the mean age at lymphoma diagnosis was 68 years (range 31–88) and the mean duration from onset of the rheumatic disease until lymphoma diagnosis was 20 years (range 1–59). The most common lymphoma subtype was DLBCL (54%), of which the non-GCB subtype dominated (71%). At the time of lymphoma diagnosis, 39% of the RA-DLBCL patients received active RA treatment defined as a disease modifying anti-rheumatic drug (DMARD) and/or corticosteroids. Twenty-three percent of the RA-DLBCL patients received a DMARD. None were treated with rituximab or any other biological DMARD during the course of RA. Detailed information about the different DMARDs used is presented in Appendix A. Most RA patients without active RA treatment at lymphoma diagnosis received non-steroidal anti-inflammatory drugs (NSAIDs). 

### 3.2. PD-1, PD-L1, PD-L2 and Lymphoma Subtypes

PD-1 was rarely expressed in tumor cells in patients with a pre-existing rheumatic disease (four cases) as well as in DLBCL controls (two cases). PD-1+ TILs were most frequent in FL (median 236 cells/HPF). In cHL, the median proportional expression of PD-1 by TILs was 9%. The highest median proportions of PD-L1+ tumor cells and TILs were observed in cHL (16% and 20%, respectively) and DLBCL (2% and 10%, respectively). PD-L2 expression was rare in tumor cells (nine cases) and generally low in TILs in most lymphoma subtypes (Figure 3A–C and Appendix A).

### 3.3. Clinical and Lymphoma Characteristics in RA-DLBCL

Clinical and lymphoma-related characteristics of the 103 RA-DLBCL cases and the 74 DLBCL controls without rheumatic disease are shown in Table 2. In RA-DLBCL patients, the highest cumulative RA-related disease activity was significantly more common in patients with high expression of PD-L1 in tumor cells compared to patients with low expression (45% vs. 20%, *p* = 0.04) (Table 3). There were significantly more women in the group with the highest RA-disease activity but no difference was seen regarding the proportion of patients with high tumor PD-L1 expression between men and women (Table 3). There were no significant differences with respect to other clinicopathological variables between the RA patients with the highest disease activity and the group with less severe RA, including EBV status (Table 4). High PD-L1 expression in tumor cells was significantly associated with EBV-positivity and high numbers of PD-1+ TILs were associated with active RA treatment at lymphoma diagnosis, while no other significant associations between expression of PD-1 or PD-L1 and clinicopathological variables were observed (Table 3). 

### 3.4. PD-1, PD-L1, PD-L2 and Overall Survival in RA-DLBCL and DLBCL Controls

Seventy-four RA-DLBCL patients had received active lymphoma treatment and were included in the survival analyses. The median OS after lymphoma diagnosis in actively treated patients was 10.3 months compared to 6.8 months for the entire group. RA-DLBCL patients with high expression of PD-L1 in tumor cells had shorter OS compared to the patients with low PD-L1 expression in tumor cells (Figure 4) (univariate HR of 2.43 (95% CI, 1.07–5.50)). High tumor cell expression of PD-L1 remained statistically significant in the multivariable analysis (HR 4.62, 95% CI, 1.55–13.71), including all variables of statistical significance in the univariate survival analyses. In the RA-DLBCL group, a high number of PD-1+ TILs was associated with superior OS in univariate analysis (HR 0.58, 95% CI, 0.35–0.98), but lost significance in the multivariable analysis. In the DLBCL controls, the median OS was 40 months, and high proportions of PD-L1+ TILs or PD-L2+ tumor cells and TILs did not affect OS in patients with RA-DLBCL or DLBCL controls (Table 5).

### 3.5. Comparison of PD-1, PD-L1 and PD-L2 in DLBCLs

When comparing the RA- and SLE-DLBCLs with DLBCL controls, we found no significant differences regarding PD-1 in TILs or PD-L1 and PD-L2 in tumor cells (Table 6). High expression of PD-L1 and PD-L2 in TILs was less common in RA-DLBCL patients compared to controls (51% vs. 77%, *p* < 0.001 and 23% vs. 74%, *p* < 0.001, respectively). A similar trend for PD-L1 in TILs was seen in SLE. A high proportion of PD-L2+ TILs was significantly more common in SLE-DLBCL compared to RA-DLBCL (92% vs. 24%, *p* < 0.001) and close to significantly more common compared to DLBCL controls (92% vs. 74% *p* = 0.057). 

## 4. Discussion

This population-based study was the first to investigate the tumor expression of PD-1, PD-L1 and PD-L2 in a cohort of patients with pre-existing rheumatic diseases and subsequent lymphoma development. High expression of PD-L1 in tumor cells in RA-DLBCL patients was more common in the RA group with the highest cumulative disease activity, indicating an association between tumor PD-L1 expression and the severity of the rheumatic disease. 

### 4.1. RA Disease Severity and PD-L1 Expression in RA-DLBCL 

Besides RA disease activity, no other significant difference was seen between the RA-DLBCL patients with the most severe RA and the lower disease activity group regarding analyzed factors possibly affecting PD-L1 expression. Lymphoma stage, EBV status and RA treatment were similar between the groups (Table 4). The proportion of patients treated with DMARDs and/or corticosteroids did not differ. Only 23 (22%) of all RA-DLBCL patients were treated with a DMARD, several different types were used, and the number of patients treated with each single drug was small. This makes a more significant impact of any individual DMARD than others on the PD-L1 expression in the study group unlikely. The use of proresid was numerically more common in the lower disease activity group, whereas antimalarials were significantly more common in the highest disease activity group, but the numbers of patients were too few to draw any conclusions from this finding (Appendix A). As previously described, assessment of RA disease activity was based on repeated measures of joint activity, ESR and the physicians´ judgement of global disease activity, reflecting the cumulative inflammatory burden of the rheumatic disease from onset until lymphoma diagnosis. In a previous study including 51 RA patients, the level of PD-L1 expression on synovial cells in joints of RA patients was associated with factors of more severe disease, including rheumatoid factor positivity, higher C-reactive protein and higher synovitis score (*p* < 0.001) [36], which support a correlation between RA disease activity and PD-L1 expression. Possibly, RA-associated proinflammatory cytokines such as IFNγ [23], TNF-*α* [21] and IL-6 [22], which are also involved in the regulation of PD-L1 [24], might contribute to higher expression of PD-L1 in the group with the most severe RA disease. Whether DLBCL patients with severe RA might be a subgroup of patients benefitting from PD-1 inhibitors needs to be addressed by future studies.

### 4.2. Expression of PD-1 and PD-L1 in Relation to Clinicopathological Parameters in RA-DLBCL

High PD-L1 expression in RA-DLBCL tumor cells was associated with EBV-positivity, consistent with findings in previous studies of DLBCL [37,38], and other EBV-associated malignancies including cHL and posttransplant lymphoproliferative disorders [39]. Active RA treatment at lymphoma diagnosis was associated with higher numbers of PD-1+ TILs in the RA-DLBCL group, but no difference was seen regarding PD-1+ TILs between the groups treated with either DMARDs or corticosteroids in monotherapy or both in combination. A previous study in RA patients (*n* = 44) found an association between increased accumulation of PD-1+ T-cells in synovial fluid and higher inflammatory joint activity (*p* = 0.043) [40]. Supposedly, patients received RA treatment due to a more severe RA, which could possibly affect PD-1 expression in TILs, although the proportion of patients belonging to the highest disease activity group did not differ between the treated and untreated cases. 

### 4.3. PD-1, PD-L1 and Overall Survival in RA-DLBCL and Comparison with Previous Studies

In RA-DLBCL patients that received active lymphoma treatment, high expression of PD-L1 in tumor cells was associated with inferior OS in multivariable analysis, whereas high expression of PD-1 in TILs was associated with superior OS in univariate analysis. Age ≥ 60 years and Ann Arbor stage III–IV were associated with inferior OS in the multivariable analysis, but no significant correlation was seen between these variables and high expression of PD-1 or PD-L1 in RA-DLBCL. High tumor cell expression of PD-L1 might be associated with inferior survival in DLBCL due to immune evasion of the tumor cells, and high numbers of PD-1+ TILs in DLBCL might reflect an immunologically active TME, which is prone to respond to chemotherapy. However, the findings from the survival analyses should be interpreted cautiously, mainly since the patients did not receive modern treatment regimens and survival was markedly poor. In the hitherto largest study of PD-1 and PD-L1 in DLBCL by Kiyasu et al., including 1253 DLBCL patients, high tumor cell expression of PD-L1 was associated with inferior OS (*p* = 0.0009), while PD-1+ TILs were not associated with OS [37]. 

### 4.4. PD-1 and PD-L1 in RA-DLBCL, SLE-DLBCL and DLBCL Controls 

The expression of PD-1 in TILs and PD-L1 in tumor cells was similar in patients with pre-existing RA, SLE and in DLBCL controls, whereas the expression of PD-L1 in TILs was significantly higher in the DLBCL controls compared to the RA-DLBCL group and close to significantly higher than the SLE-DLBCL group. These findings are somewhat unclear and might be explained by an altered inflammatory tumor milieu in DLBCL patients with RA or SLE. In addition, we cannot rule out that these findings may be related to the fact that the RA-DLBCL material was older than the DLBCL controls, since the age of FFPE material may influence the results of the immunohistochemical markers [41]. However, when older (diagnosed 1965–1986) versus newer (diagnosed 1987–2005) RA-DLBCL cases were compared, no differences were found in expression of PD-1 or PD-L1. 

### 4.5. Comparison of PD-1, PD-L1 and PD-L2 between Various Lymphoma Subtypes

Among the various included lymphoma subtypes, the expression of PD-L1 in tumor cells and TILs was highest in cHL. PD-L1 is commonly overexpressed by HRS cells due to frequent genetic alterations in the chromosome 9p24.1 region containing the PD-L1 locus [42] and both PD-1+ and PD-L1+ TILs frequently occur in the TME. Median expression of PD-L1 in TILs, PD-L1 in HRS cells and PD-1 in TILs were higher in the present study compared to a previous study from our research group where 387 cases with cHL were included irrespective of pre-existing rheumatic diseases (20%, 16% and 9% vs. 12%, 0% and 2%, respectively) [3]. Our findings suggest a generally higher expression of PD-1 and PD-L1 in cHL patients with pre-existing rheumatic diseases, possibly induced by a higher inflammatory activity in patients with rheumatic diseases. However, due to the low number of cases in the present study, these findings should be interpreted with great caution. Among the NHL subtypes, DLBCL had the highest expression of PD-L1 in both tumor cells and TILs. Expression of PD-1 in TILs was highest in FL followed by cHL and DLBCL. This is consistent with results from previous studies showing PD-1 expression at high levels on germinal center follicular cells in FL [43]. 

### 4.6. Strengths and Weaknesses

Strengths of our study include the population-based setting with histopathological review of the cases, a large study group in the context of lymphoma developed in patients with a pre-existing rheumatic disease and the detailed clinical information of the rheumatic diseases including a cumulative disease activity score for RA-DLBCL cases. Furthermore, the high proportion of patients untreated for their rheumatic disease reduces the risk of anti-rheumatic drugs affecting the results. Our digital image analysis approach and the use of PD-ligand/PAX-5 double staining technique allowed a reliable and consistent identification of tumor cells (PAX-5 positive) and leukocytes (PAX-5 negative). Weaknesses include the fact that most of the cases with pre-existing rheumatic diseases were diagnosed with lymphoma before the 1990s, making comparison with the more recent control group difficult and in particular affecting the reliability of the survival analyses.

## 5. Conclusions

In conclusion, our study is the first to investigate the expression of PD-1, PD-L1 and PD-L2 in lymphoma patients with a pre-existing rheumatic disease. Our results indicate an association between a more severe RA disease and increased expression of PD-L1 in DLBCL tumor cells. This is of mechanistic pathophysiological interest and might also have implications for the prognosis of the lymphoma. Whether DLBCL patients with severe RA might be a subgroup of patients benefitting from PD-1 inhibitors needs to be addressed by future studies. The findings warrant further studies also in other chronic inflammatory conditions in order to elucidate whether this association is restricted to RA-associated DLBCL or if a more general mechanism related to chronic inflammation could be suspected.

## Figures and Tables

**Figure 1 cancers-14-01509-f001:**
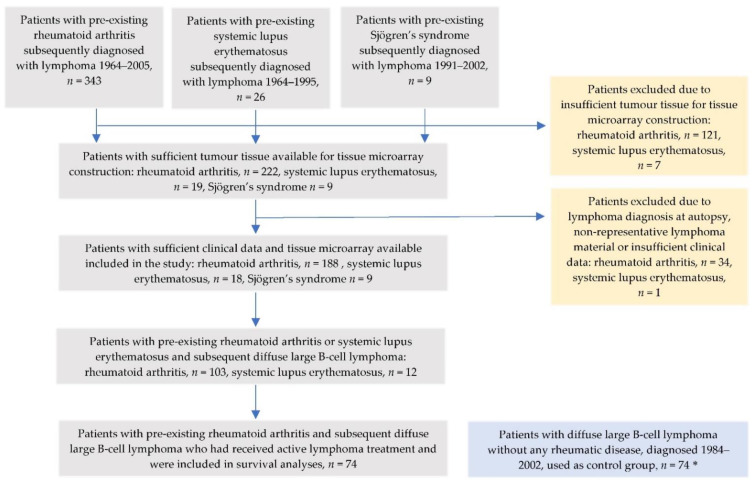
Flowchart of patients with pre-existing rheumatoid arthritis, systemic lupus erythematosus or Sjögren’s syndrome and subsequent lymphoma development included in the study with complete clinical information and with sufficient tumor tissue for tissue microarray construction. * Of 85 originally identified DLBCL patients without any rheumatic disease, specimens suitable for TMA construction remained in 74 patients used as controls for comparison with the rheumatoid arthritis- and systemic lupus erythematosus-DLBCL cases.

**Figure 2 cancers-14-01509-f002:**
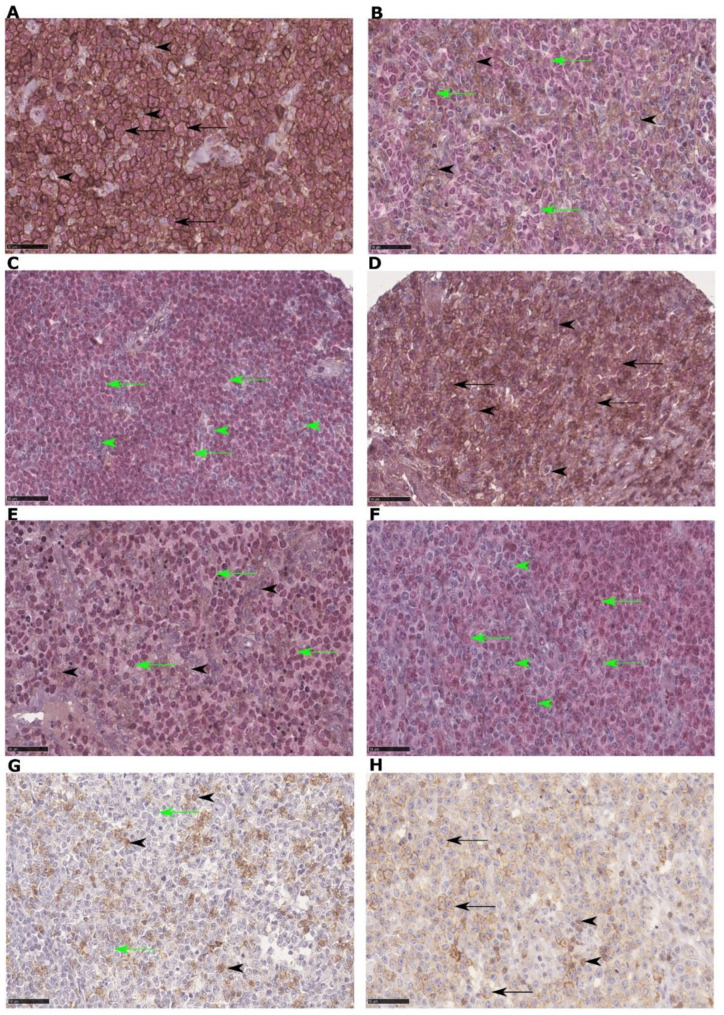
Patients with rheumatoid arthritis with diffuse large B-cell lymphoma stained with PD-1, PD-L1/PAX5 and PD-L2/PAX5 (antibody NAT105/ab52587 (Abcam, Cambridge, UK) for PD-1, antibody E1L3N/13684 (Cell Signaling Technology, Danvers, MA, USA) for PD-L1, antibody D7U8C/82723 (Cell Signaling Technology) for PD-L2 and antibody M7307/DAK-Pax5 (Dako, Santa Clara, CA, USA) for PAX5). (**A**): High amounts of PD-L1+ tumor cells and TILs. (**B**): A high amount of PD-L1+ TILs and a low amount of PD-L1+ tumor cells. (**C**): Low amounts of PD-L1+ tumor cells and TILs. (**D**): High amounts of PD-L2+ tumor cells and TILs. (**E**): A high amount of PD-L2+ TILs and a low amount of PD-L2+ tumor cells. (**F**): Low amounts of PD-L2+ tumor cells and TILs. (**G**): A high amount of PD-1+ TILs and no PD-1+ tumor cells. (**H**): PD-1+ tumor cells and a few scattered PD-1+ TILs. Magnification: 400× in all images. Legends: PD-1+/PD-L1+/PD-L2+ tumor cells = black arrow, PD-1+/PD-L1+/PD-L2+ TILs = black arrowhead, PD-1-/PD-L1-/PD-L2- tumor cells = green arrow, PD-1-/PD-L1-/PD-L2- TILs = green arrowhead.

**Figure 3 cancers-14-01509-f003:**
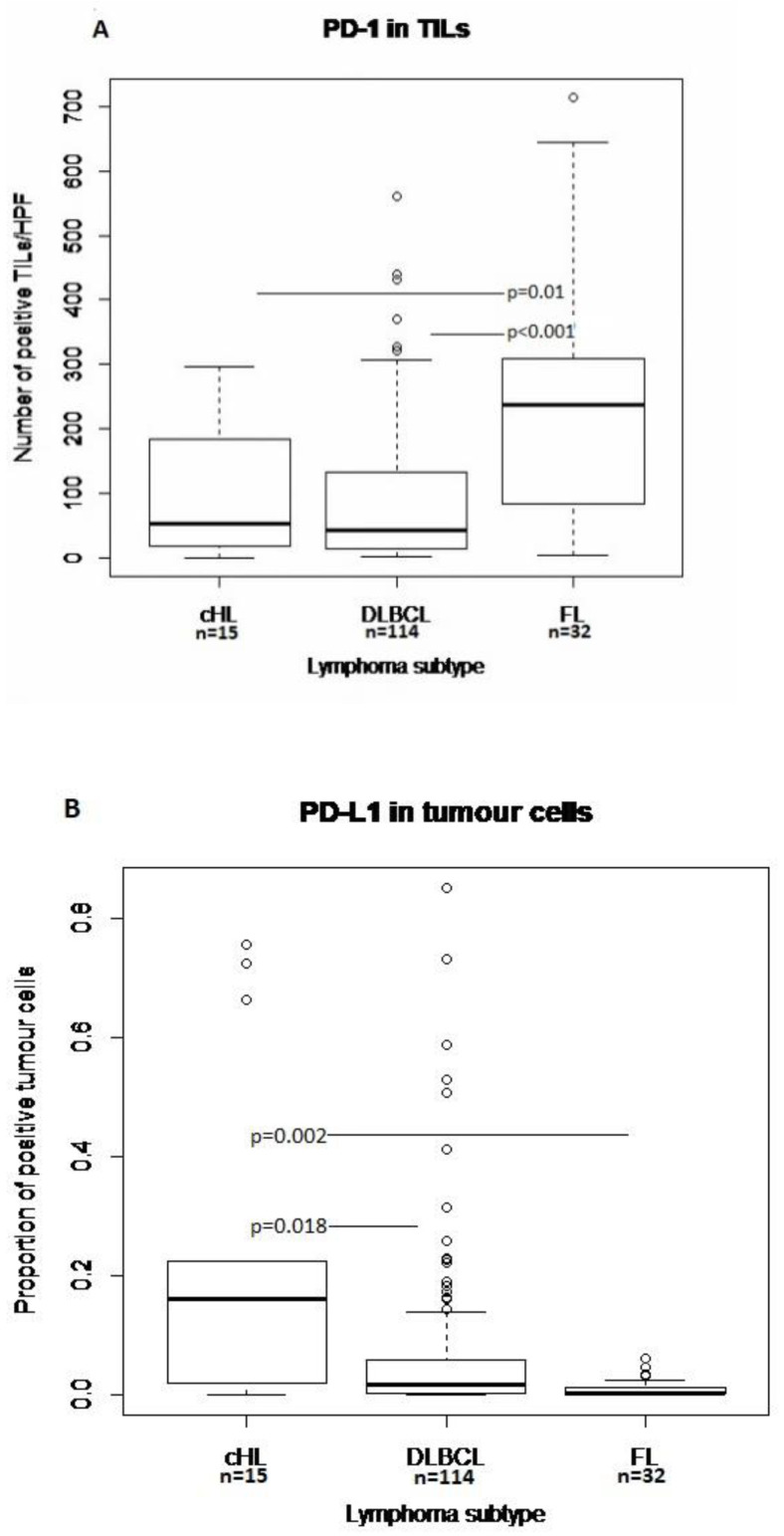
Boxplots of distribution of proportions of PD-L1+ tumor cells (**A**) and TILs (**B**) and number of PD-1+ TILs/HPF (**C**) in cHL, DLBCL and FL. Comparison of median values with the group with the highest value, *p*-values according to Wilcoxon signed rank-test. TILs, tumor infiltrating leukocytes; cHL, classical Hodgkin lymphoma; DLBCL, diffuse large B-cell lymphoma; FL, follicular lymphoma.

**Figure 4 cancers-14-01509-f004:**
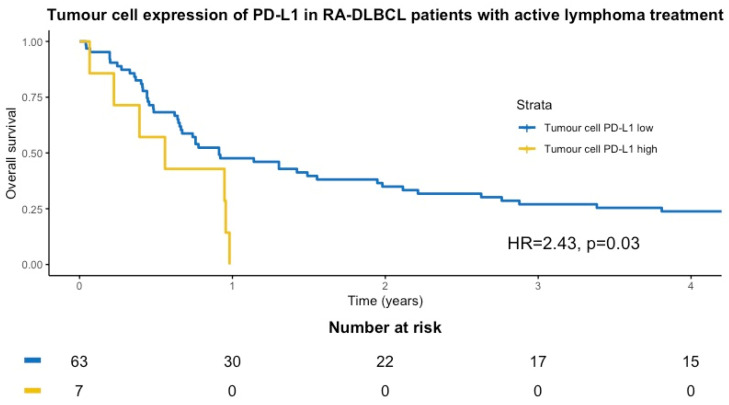
Kaplan–Meier curve according to ≥17% (yellow line) and <17% (blue line) PD-L1+ tumor cells in patients with rheumatoid arthritis and diffuse large B-cell lymphoma.

**Table 1 cancers-14-01509-t001:** Lymphoma subtypes and clinical characteristics of 215 patients with a pre-existing rheumatic disease.

	Entire Cohort	cHL	DLBCL	FL	LPL	MZBL	CLL	HGBCL	LGBCL	MCL	TCL	Burkitt Lymphoma
All patients, *n* (%)	215 (100)	16 (100)	116 (100)	32 (100)	5 (100)	5 (100)	13 (100)	8 (100)	7 (100)	5 (100)	5 (100)	3 (100)
Age at lymphoma diagnosis in years: mean (range)	68.0 (31–88)	63.4 (46–80)	68.9 (31–87)	66.8 (46–88)	70.4 (57–82)	49.8 (35–67)	69.7 (43–87)	71.4 (57–80)	74.6 (68–80)	70.6 (56–78)	64.8 (48–73)	67.0 (59–75)
Duration of rheumatic disease until lymphoma diagnosis in years: mean (range)	19.7 (1–59)	12.1 (1–33)	19.3 (2–50)	21.3 (2–53)	7.3 (4–10)	14.3 (2–33)	24.4 (4–50)	23.5 (3–52)	30.3 (10–59)	21.8 (13–26)	37.0 (37)	22.0 (7–47)
Missing, *n* (%)	18 (8)	2 (13)	1 (1)	4 (13)	1 (20)	2 (40)	2 (15)	0 (0)	1 (14)	1 (20)	4 (80)	0 (0)
Sex, *n* (%)												
Women	134 (62)	8 (50)	74 (64)	19 (60)	2 (40)	5 (100)	9 (69)	7 (88)	2 (29)	3 (60)	4 (80)	1 (33)
Men	81 (38)	8 (50)	42 (36)	13 (40)	3 (60)	0 (0)	4 (31)	1 (12)	5 (71)	2 (40)	1 (20)	2 (67)
Ann Arbor stage, *n* (%)												
I–II	69 (32)	5 (31)	37 (32)	15 (47)	1 (20)	4 (80)	1 (8)	3 (37)	3 (43)	0 (0)	0 (0)	0 (0)
III–IV	123 (57)	8 (50)	74 (64)	13 (41)	2 (40)	1 (20)	10 (77)	5 (63)	3 (43)	4 (80)	1 (20)	2 (67)
Missing	23 (11)	3 (19)	5 (4)	4 (13)	2 (40)	0 (0)	2 (15)	0 (0)	1 (14)	1 (20)	4 (80)	1 (33)
EBV status, *n* (%)												
Positive	22 (10)	7 (44)	10 (9)	0 (0)	2 (40)	1 (20)	0 (0)	2 (25)	0 (0)	0 (0)	0 (0)	0 (0)
Negative	182 (84)	9 (66)	103 (89)	30 (94)	2 (40)	3 (60)	11 (85)	5 (63)	7 (100)	4 (80)	4 (80)	3 (100)
Missing	12 (6)	0 (0)	3 (3)	2 (6)	1 (20)	1 (20)	2 (15)	1 (13)	0 (0)	1 (20)	1 (20)	0 (0)
Rheumatic disease, *n* (%)												
RA	188 (87)	15 (94)	103 (89)	30 (94)	3 (60)	2 (40)	11 (85)	7 (88)	7 (100)	5 (100)	2 (40)	3 (100)
SLE	18 (8)	1 (6)	12 (10)	1 (3)	1 (20)	0 (0)	0 (0)	0 (0)	0 (0)	0 (0)	3 (60)	0 (0)
PSS	9 (4)	0 (0)	1 (1)	1 (3)	1 (20)	3 (60)	2 (15)	1 (12)	0 (0)	0 (0)	0 (0)	0 (0)

EBV, Epstein–Barr virus; RA, rheumatoid arthritis; SLE, systemic lupus erythematosus; PSS, primary Sjögren´s syndrome; cHL, classical Hodgkin lymphoma; DLBCL, diffuse large B-cell lymphoma; FL, follicular lymphoma; CLL, chronic lymphocytic leukemia; LPL, lymphoplasmacytic lymphoma; MZBL, marginal zone B-cell lymphoma; HGBCL, high-grade B-cell lymphoma; LGBCL, low-grade B-cell lymphoma; MCL, mantle cell lymphoma; TCL, T-cell lymphoma.

**Table 2 cancers-14-01509-t002:** Clinical and lymphoma-related characteristics of 103 RA-DLBCL cases and 74 DLBCL controls without rheumatic disease.

	RA-DLBCL, *n* = 103	DLBCL Controls, *n* = 74	*p*-Value **
Sex, *n* (%)			0.12
Women	61 (59)	35 (47)	
Men	42 (41)	39 (53)	
Age at lymphoma diagnosis in years:			0.75 ***
median (range)	71 (31–87)	71 (32–90)	
Age ≥ 60 years, *n* (%)			0.37
Yes	85 (83)	57 (77)	
No	18 (17)	17 (33)	
Decade of lymphoma diagnosis, *n* (%)			<0.001
1990s–2000s	34 (33)	66 (85)	
1960s–1980s	69 (67)	11 (15)	
Survival after lymphoma diagnosis in months, all RA-DLBCL			<0.001 ***
Median (range)	6.8 (0–376)	40 (0.5–253)	
Missing, *n* (%)	2 (2)	0 (0)	
DLBCL subtype, *n* (%)			0.008
GCB	30 (29)	35 (47)	
non-GCB	69 (67)	34 (46)	
Missing	4 (4)	5 (7)	
Ann Arbor stage, *n* (%)			0.26
I–II	34 (33)	30 (41)	
III–IV	65 (63)	42 (57)	
Missing	4 (4)	2 (3)	
B-symptoms, *n* (%)			0.02
Yes	21 (20)	30 (41)	
No	62 (61)	39 (54)	
Missing	20 (19)	4 (5)	
Extranodal involvement *, *n* (%)			0.03
Yes	58 (57)	29 (39)	
No	43 (41)	42 (57)	
Missing	2 (2)	3 (4)	
Bone marrow involvement, *n* (%)			NA
Yes	15 (15)	NA	
No	83 (81)	NA	
Missing	5 (5)	NA	
Increased LDH, *n* (%)			0.55
Yes	20 (19)	40 (54)	
No	8 (8)	22 (30)	
Missing	75 (73)	12 (16)	
EBV status, *n* (%)			NA
Positive	9 (9)	NA	
Negative	92 (91)	NA	
Missing	2 (2)	NA	
Lymphoma treatment, *n* (%)			<0.001
No active treatment	25 (24)	0 (0)	
CT +/- radiotherapy	55 (54)	67 (96)	
Radiotherapy	12 (12)	3 (0)	
Surgery	2 (2)	1 (0)	
Surgery + radiotherapy	6 (6)	0 (0)	
Missing	4 (4)	3 (4)	

RA, rheumatoid arthritis; DLBCL, diffuse large B-cell lymphoma; GCB, germinal center B-cell like; LDH, lactate dehydrogenase; EBV, Epstein–Barr virus; CT, chemotherapy; NA, not available. * Bone marrow involvement not included. ** According to chi-square or Fischer’s exact test, unless otherwise indicated. *** According to Wilcoxon signed rank-test.

**Table 3 cancers-14-01509-t003:** Clinical and lymphoma-related parameters in the RA-DLBCL patients according to expression of PD-L1 in tumor cells and TILs and PD-1 in TILs.

	Entire Cohort	PD-L1 in Tumor Cells ≥ 17%	PD-L1 in Tumor Cells < 17%	*p*-Value *	PD-L1 in TILs ≥ 10%	PD-L1 in TILs < 10%	*p*-Value *	PD-1 in TILs ≥ 21/HPF	PD-1 in TILs < 21/HPF	*p*-Value *
All patients, *n* (%)	103 (100)	11 (100)	89 (100)		51(100)	49 (100)		64 (100)	35 (100)	
Sex, *n* (%)				0.70			0.14			0.43
Women	61 (59)	6 (55)	54 (61)		27 (53)	33 (67)		40 (63)	19 (54)	
Men	42 (41)	5 (45)	35 (39)		24 (47)	16 (33)		24 (38)	16 (46)	
Age in years, *n* (%)				0.68			0.93			0.28
≥60	85 (83)	10 (91)	72(81)		42 (82)	40 (82)		50 (78)	31 (89)	
<60	18 (19)	1(9)	17 (19)		9 (18)	9 (18)		14 (22)	4 (11)	
DLBCL subtype, *n* (%)				0.35			0.19			0.27
GCB	30 (29)	2 (18)	27 (30)		12 (24)	17 (35)		21 (33)	8 (23)	
Non-GCB	69 (67)	9 (82)	57 (64)		37 (73)	29 (59)		40 (63)	26 (74)	
Missing	4 (4)	0 (0)	5 (6)		2 (4)	3 (6)		3 (4)	1 (3)	
Ann Arbor stage, *n* (%)				0.31			0.12			0.21
I–II	34 (33)	5 (45)	29 (33)		21 (41)	13 (27)		25 (39)	9 (26)	
III–IV	65 (63)	5 (45)	57 (64)		28 (55)	34 (69)		37 (58)	24 (68)	
Missing	4 (4)	1 (10)	3 (3)		2 (4)	2 (4)		2 (3)	2 (6)	
EBV status, *n* (%)				<0.001			0.11			0.88
Positive	9 (9)	4 (36)	5 (6)		7 (14)	2 (4)		6 (9)	3 (9)	
Negative	92 (89)	7 (64)	82(92)		44 (86)	45 (92)		57 (89)	32 (91)	
Missing	2 (2)	0 (0)	2 (2)		0 (0)	2 (4)		1 (2)	0 (0)	
RA highest disease severity group, *n* (%)				0.04			0.81			0.84
Yes	25 (25)	5 (45)	18 (20)		12 (23)	11 (22)		15 (23)	9 (26)	
No	74 (72)	5 (45)	68 (77)		36 (71)	37 (76)		46 (72)	25 (71)	
Missing	4 (4)	1 (10)	3 (3)		3 (6)	1 (2)		3 (5)	1 (3)	
Active RA treatment, *n* (%) **				0.96			0.84			0.02
Yes	39 (38)	4 (36)	34 (38)		20 (39)	18 (37)		29 (45)	8 (23)	
No	61 (59)	6 (55)	53 (60)		29 (57)	30 (61)		32 (50)	27 (77)	
Missing	3 (3)	1 (10)	2 (2)		2 (4)	1 (2)		3 (5)	0 (0)	
Type of RA treatment, *n* (%)				0.66			0.76			0.46
Corticoteroids monotherapy	16 (16)	2 (18)	14 (16)		9 (18)	7 (14)		10 (16)	5 (14)	
DMARD monotherapy	14 (14)	2 (18)	12 (13)		6 (12)	8 (16)		12 (19)	2 (6)	
Corticosteroids + DMARD	9 (9)	0 (0)	8 (9)		5 (10)	3 (6)		7 (11)	1 (3)	
Year of lymphoma diagnosis, *n* (%)				0.89			0.31			0.09
<1987	49 (48)	5 (45)	43 (48)		27 (53)	21 (43)		27 (42)	21 (60)	
≥1987	54 (52)	6 (55)	46 (52)		24 (47)	28 (57)		37 (58)	14 (40)	

RA, rheumatoid arthritis; DLBCL, diffuse large B-cell lymphoma; GCB, germinal center B-cell like; EBV, Epstein–Barr virus; DMARD, disease modifying anti-rheumatic drug; TIL, tumor-infiltrating leukocytes. * According to chi-square or Fischer’s exact test. ** Active RA treatment = DMARD and/or corticosteroids.

**Table 4 cancers-14-01509-t004:** RA- and lymphoma-related characteristics in DLBCL of the highest RA disease activity group vs. the lower disease activity group.

	RA Highest Disease Activity Group	RA Lower Disease Activity Group	*p*-Value *
Entire group, *n* (%)	25 (100)	74 (100)	
Sex, *n* (%)			
Women	19 (76)	39 (53)	0.04
Men	6 (24)	35 (47)	
Year of lymphoma diagnosis: median (range)	1987 (1967–1995)	1987 (1966–1997)	0.67 **
Age at lymphoma diagnosis: median (range)	71 (47–84)	71 (31–87)	0.59 **
Age in years, *n* (%)			0.75
≥60	22 (88)	62 (84)	
<60	3 (12)	12 (16)	
DLBCL subtype, *n* (%)			0.28
GCB	5 (14)	24 (30)	
non-GCB	18 (78)	47 (66)	
Missing	2 (8)	3 (4)	
Ann Arbor stage, *n* (%)			0.57
I–II	7 (28)	24 (29)	
III–IV	18 (72)	46 (66)	
Missing	0 (0)	4 (5)	
EBV status, *n* (%)			0.99
Positive	2 (8)	6 (8)	
Negative	22 (88)	67 (91)	
Missing	1 (4)	1 (1)	
Active RA treatment, *n* (%) ***			0.94
Yes	10 (40)	29 (39)	
No	15 (60)	45 (61)	
Any DMARD, *n* (%)			0.92
Yes	6 (24)	17 (23)	
No	19 (76)	72 (77)	
Corticosteroids for RA, *n* (%)			0.48
Yes	5 (20)	20 (27)	
No	20 (80)	54 (73)	

RA, rheumatoid arthritis; DLBCL, diffuse large B-cell lymphoma; GCB, germinal center B-cell like; EBV, Epstein–Barr virus; DMARD, disease modifying anti-rheumatic drugs. * According to chi-square or Fischer’s exact test, unless otherwise indicated. ** According to Wilcoxon signed rank-test. *** Active RA treatment = DMARD and/or corticosteroids.

**Table 5 cancers-14-01509-t005:** Relative risk of death from any cause estimated as HRs with 95% CIs and *p*-values in RA- DLBCL patients exposed to any lymphoma treatment and DLBCL controls by putative prognostic factors.

	RA-DLBCL		DLBCL Controls			
Exposure	N	Univariate	Multivariable	N	Univariate	Multivariable
tPD-L1 ≥ 17%	70	2.43: 1.07–5.50, 0.03	4.62: 1.55–13.71, 0.006	74	1.28: 0.68–2.41, 0.50	-
lPD-L1 ≥ 10%	70	1.46: 0.90–2.37, 0.10	-	74	0.69: 0.40–1.20, 0.20	-
lPD-1 ≥ 21/HPF	69	0.58: 0.35–0.98, 0.04	0.82: 0.45–1.50, 0.52	73	0.40: 0.23–0.69, 0.001	0.41: 0.25–0.80, 0.007
tPD-L2 ≥ 8%	70	0.53: 0.19–1.47, 0.23	-	74	0.63: 0.23–1.73, 0.40	-
lPD-L2 ≥ 2%	70	0.80: 0.46–1.39, 0.42	-	74	0.95: 0.56–1.63, 0.90	-
Age in years ≥ 60	73	3.48: 1.69–7.17, <0.001	3.16: 1.45–6.90, 0.004	74	2.04: 1.16–3.60, 0.01	1.79: 0.98–3.28, 0.06
Male	73	1.50: 0.92–2.44, 0.10	-	74	1.32: 0.83–2.10, 0.20	-
Ann Arbor stage III-IV	73	2.70: 1.58–4.60, <0.001	3.10: 1.55–6.19, 0.001	74	2.47:1.50–4.08, <0.001	1.97: 1.09–3.57, 0.03
B symptoms	67	1.65: 0.89–3.06, 0.11	-	70	1.78: 1.09–2.91, 0.02	1.49: 0.81–2.71, 0.20
Non-GCB	68	1.82: 1.06–3.11, 0.03	1.85: 0.999–3.43, 0.0502	69	2.00: 1.23–3.28, 0.006	1.78: 1.05–3.04, 0.03
EBV positive	71	2.57: 1.00–6.63, 0.049	1.23: 0.43–3.54, 0.70	NA	-	-
RA highest disease activity group	73	1.51: 0.88–2.60, 0.14	-	NA	-	-

HR, hazard ratio; CI, confidence interval; RA, rheumatoid arthritis; DLBCL, diffuse large B-cell lymphoma; tPD-L1, PD-L1 in tumor cells; lPD-L1, PD-L1 in tumor infiltrating leukocytes; lPD-1, PD-1 positive tumor infiltrating leukocytes; EBV, Epstein–Barr virus.

**Table 6 cancers-14-01509-t006:** Comparison of PD-L1+ and PD-L2+ tumor cells and TILs and PD-1+ TILs in RA-DLBCL, SLE-DLBCL and DLBCL controls.

	RA-DLBCL	SLE-DLBCL	DLBCL Controls	*p*-Value *, RA-DLBCL vs. DLBCL Controls	*p*-Value *, SLE-DLBCL vs. DLBCL Controls	*p*-Value *, RA-DLBCL vs. SLE-DLBCL
All patients, *n* (%)	103 (100)	12 (100)	74 (100)			
tPD-L1 ≥ 17%, *n* (%)				0.26	0.38	0.12
Yes	11 (11)	3 (25)	12 (16)			
No	89 (86)	8 (67)	61 (82)			
Missing	3 (3)	1 (8)	1 (1)			
lPD-L1 ≥ 10%, *n* (%)				<0.001	0.09	0.82
Yes	51 (50)	6 (50)	57 (77)			
No	49 (48)	5 (42)	16 (22)			
Missing	3 (3)	1 (8)	1 (1)			
lPD1 ≥ 21/HPF, *n* (%)				0.19	0.47	0.98
Yes	64 (65)	7 (58)	54 (73)			
No	35 (35)	4 (33)	19 (26)			
Missing	4 (4)	1 (8)	1 (1)			
tPD-L2 ≥ 8%, *n* (%)				0.87	0.63	0.73
Yes	6 (6)	1 (8)	4 (5)			
No	94 (91)	10 (83)	70 (95)			
Missing	3 (3)	1 (8)	0 (0)			
lPD-L2 ≥ 2%, *n* (%)				<0.001	0.057	<0.001
Yes	24 (23)	11 (92)	55 (74)			
No	76 (74)	0 (0)	19 (26)			
Missing	3 (3)	1 (8)	0 (0)			

TILs, tumor infiltrating leukocytes; SLE, systemic lupus erythematosus; RA, rheumatoid arthritis; DLBCL, diffuse large B-cell lymphoma; tPD-L1, PD-L1 in tumor cells; lPD-L1, PD-L1 in tumor infiltrating leukocytes; lPD-1, PD-1 positive tumor infiltrating leukocytes. * According to chi-square or Fischer’s exact test.

## Data Availability

The data presented in this study are available upon reasonable request from the corresponding author.

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
