# Peer review of "Expression of PD-1, PD-L1 and PD-L2 in Lymphomas in Patients with Pre-Existing Rheumatic Diseases—A Possible Association with High Rheumatoid Arthritis Disease Activity"

_cancers, 2022, doi:10.3390/cancers14061509_

Round 1

Reviewer 1 Report

In the present research, authors study the expression of PD-1- PD-L1 in lymphoma patients with preexisting rheumatic disease and infer that patients with RA and B cell lymphoma could benefit from anti-PD-1 immunotherapy. However, the inference should be viewed with caution. My comments are appended as below:

  1. Line 56, reference 2-6- elaborate more on mechanistic point and annotate with statistical inference.
  2. While citing references with patients' information, annotate with a number of patients included and statistical inference.
  3. Correct figure no 1- seems superimposed with numbers.
  4. IHC- PD-L1/2 expression- does the antibody detects soluble form or membrane component? Please specify the antibody dilutions used.
  5. Positive and negative area in IHC- what was the staining cutoff used? This categorization looks subjective.
  6. Figure 2- please provide magnified image for better visualization.
  7. Figure3- include n and annotate with statistical inference.
  8. Figure 4- include HR along with P-value.
  9. Do authors observe other cofounders as diabetes and hypertension to be correlated with responsiveness among the patients? It was previously noted that a bunch of cofounders affects the immunotherapy efficacy: PMID: 33076303. The authors may note this in the discussion part.
  10. Do authors study other immune cells in the TME as MDSCs which are previously said to be associated with immunotherapy efficacy?

Reviewer 2 Report

The authors describe the expression of proteins of the PD-1 axis in a cohort of patients who developed lymphoma and had pre-existing rheumatic disease. High PD-L1 expression in tumor cells was associated with high RA activity and  EBV positivity, while high numbers of PD-1 positive TILs were associated with active RA treatment at the time of lymphoma diagnosis. Patients with high PD-L1 expression in the tumor had a worse OS. There was no difference in the percentage of PD-L1 expression in the tumor of DLBCL patients compared with a control group, and there was less PD-L1 expressed in the TILs of RA DLBCL patients compared tot he control DLBCL group.PD-L2 was expressed higher in SLE DLBCL than RA DLBCL. The research is solid, the main problem is the RA cohort was diagnosed before 2000 and the treatment is not comparable to the control DLBCL cohort, but this is addressed in the discussion.

Questions:

  1. In the introduction the connection of PD-1 treatment and the occurence of auto immune diseases used as a proof that the PD-1 axis is important in RA, I agree with that statement, but am very much suprised that it is suggested several times that these patients since they have high expression of PD-L1 could benefit from PD-1 treatment. These suggestions need to be removed or better substantiated.
  2. Is RA activity also based on the treatment the patients have recieved over the years? The treatment at the time of lymphoma diagnosis important, but the accumulation of treatment also.
  3. Treatment regimens between RA-DLBCL and control DLBCLs are very different, is it possible to compare risk of death? And do these results of the risk analysis on patients that have been treated so different mean anything today?
  4. Is the reason that there is no recent RA patients with lymphoma that RA treatment has improved or is it available data? Maybe good to look at more recent numbers and show this is not an outdated problem?

Round 2

Reviewer 1 Report

no comments